behaviour/ecology

inadvertent social information (ISI), information spread, decision-making, non-grouping animals, temporary aggregations

**Author for correspondence:**
Zoltán Tóth
e-mail: toth.zoltan@atk.hu

# Tutors do not facilitate rapid resource exploitation in temporary tadpole aggregations

## Zoltán Tóth and Boglárka Jaloveczki

Department of Zoology, Plant Protection Institute, Centre for Agricultural Research, ELKH, Budapest, Hungary

ZT, 0000-0002-2634-8393

The utilization of social cues is usually considered an important adaptation to living in social groups, but recent evidence suggests that social information use may be more prevalent in the animal kingdom than previously thought. However, it is debated whether such information can efficiently diffuse in temporary aggregations of non-grouping individuals where social cohesion does not facilitate information transmission. Here, we provide experimental evidence that a simple social cue, the movement of conspecifics in a structured environment affected individuals' spatial decisions in common frog (*Rana temporaria*) tadpoles and thereby facilitated the discovery rate of a novel food patch. However, this was true only in those tadpole collectives that consisted solely of untutored individuals. In those collectives where tutors with prior experience with the presented food type were also present, this social effect was negligible most probably due to the difference in activity between naive and tutor individuals. We also showed that the proportion of tadpoles that discovered the food patch was higher in the control than in the tutored collectives, while the proportion of feeding tadpoles was only marginally higher in the latter collectives. Our findings indicate that social information use can influence resource acquisition in temporary aggregations of non-grouping animals, but individual differences in satiety may hinder effective information spread associated with exploitable food patches.

## 1. Introduction

The presence, behaviour or product of the behaviour of others often serve as social cues for group-living animals, conveying inadvertent social information (ISI) that may be used adaptively during decision-making [1]. In such species, ISI use can affect a wide variety of individual-level behaviours including foraging

and habitat choice [2], and by inducing strong correlations in the behaviour and space use of many individuals, has the potential to mediate interactions with resources, predators and competitors at the population- or community-level [3]. In a number of group-living species, the transmission of social information has also been proved to play a crucial role in the spread of experimentally induced skills, innovations and subsequent (cultural) traditions [4–8].

Substantial evidence suggests that ISI use is not confined to species characterized by social tendencies and permanent group-living; individuals of non-grouping species have been shown to exploit social cues as well [9–12]. This raised the idea that even in loose, temporary aggregations where the social attraction between group-mates (as in [13]) does not maintain spatial proximity and group cohesion, behavioural correlations driven by social information use may emerge [3]. For instance, if the response of an individual to predation threat exerts similar behavioural adjustments in conspecific observers, such socially acquired information may diffuse among nearby individuals and generate behavioural responses beyond the detection range of the original cue [14]. Such diffusion can occur through the network of connections that represent the number of opportunities an individual has to observe the behaviour of others [15,16], i.e. observations based on visual and/or non-visual (e.g. chemical, acoustic or vibration-related) perception, and may facilitate predator avoidance or resource localization/acquisition in various species and ecological scenarios [17–19].

In a foraging context, animals living in cohesive groups often rely on social information when they have to make decisions about where to feed, and by doing so, individuals can reduce the latency of discovering alternative food patches [20–23]. Social cues produced by experienced group-mates are also used to overcome initial neophobia when animals are confronted with a novel food type [24,25]. In non-grouping animals, the behaviour of conspecific or heterospecific tutors has also been found to facilitate individuals' foraging performance. For instance, juvenile Port Jackson sharks (*Heterodontus portusjacksoni*) were more successful in a novel foraging task when paired with a trained conspecific compared to individual foragers or those in the presence of a sham demonstrator [26]. Similarly, individuals of two shoaling and four non-grouping fish species were equally capable of exploiting social cues that were provided by groups of shoal-forming heterospecifics during a foraging task [11]. In principle, social information may diffuse and mediate the spatial and foraging decisions of multiple individuals when animals aggregate to exploit some patchily distributed resource (e.g. [12]). Relative differences in activity and associated conspicuousness among individuals may be a simple but suitable mechanism that could facilitate naive observers' access to alternative resources when individuals with prior experience are present. For instance, in wood frog (*Rana sylvatica*) tadpoles, tutors that reduced their activity to a greater extent compared to the baseline level in the presence of predator cues conveyed information about predation risk to others more efficiently than those tutors that changed their activity less radically [27].

In this study, we investigated how individuals use social cues in a foraging task in collectives of common frog (*Rana temporaria*) tadpoles. During 40 min long trials, individuals had to locate a new food patch of high nutritional value (a *Spirulina* tablet) in an unfamiliar, structured arena and feed from this resource. Control (C) collectives consisted of eight naive individuals, while the *Spirulina* (S) collectives consisted of five naive and three *Spirulina*-treated tadpoles. These latter individuals were supplemented with chemical cues of *Spirulina* during their ontogenetic development, so they served as potential social cue producers in their collectives during the trials. We also used a neutral cue (snail water) that these tadpoles could associate with *Spirulina* and expected that in its presence the conditioned tadpoles will search for food more actively at test (associative learning ability has been proven in tadpoles in various contexts in several frog species (e.g. [28,29]). We predicted that if the localization and exploitation of an unknown food source can be socially mediated, then the relative activity, spatial movement and foraging behaviour (including reduced neophobia towards the food patch) of *Spirulina*-treated individuals should decrease the time required for naive tadpoles to feed from the presented food patch in the S collectives.

## 2. Material and methods

### 2.1. Study subjects

We collected common frog (*Rana temporaria*) eggs from eight egg clutches from a natural breeding pond (47°44′35″ N, 19°0′09″ E) located in the Pilis-Visegrádi Mountains, Hungary on 18 March 2019 and transported them to the laboratory of the Plant Protection Institute, Centre for Agricultural Research.

During collection, we sampled clutches that were laid at the periphery of the mass spawn; however, egg clutches may be fertilized by more than one male [30], so there was some chance that certain tadpoles from different clutches could be related. On the other hand, the tested collectives (see below) consisted of one randomly selected individual per clutch, so the potential presence of kin might have affected the two types of collectives with an equal probability. After hatching, we randomly selected 40–40 individuals from each clutch and allocated them in groups of eights into five plastic containers, each containing approximately 6 l reconstituted soft water (RSW; 48 mg l$^{-1}$ NaHCO$_3$, 30 mg l$^{-1}$ CaSO$_4 \times$ 2H$_2$O, 61 mg l$^{-1}$ MgSO$_4 \times$ 7H$_2$O and 2 mg l$^{-1}$ KCl dissolved in reverse-osmosis filtered tap water and treated with UV). We arranged the containers with the control and *Spirulina*-treated individuals in four rows on the shelves in the laboratory and assigned their positions at random. We fed the tadpoles ad libitum with slightly boiled, chopped spinach and changed the water in their containers twice a week. Great pond snails (*Lymnaea stagnalis*) were collected from a small artificial pond near the institute (47°33′04″ N, 18°55′36″ E). Sixteen animals were housed in a container containing 8 l RSW and equipped with two clay flower pots, while additional 8–8 snails were kept in two containers, each containing 4 l RSW and a clay pot (i.e. 2 snails l$^{-1}$ in each container). Similarly to the tadpoles, snails were also fed with spinach ad libitum (body mass (mean ± s.d.): 4.87 ± 1.57 g, $n = 32$). Water from the containers of these snails (snail water) was used as an associative cue during the ontogenetic treatment and in the trials. We set room temperature in the laboratory to 20°C during daylight hours and allowed the temperature to drop during night. Lighting was set to a 11.75 : 12.25 h light : dark cycle in the beginning, then day length was increased by half an hour in accordance with the projected calendar day length (every 7–10 days) to simulate natural changes in the photoperiod. After completion of the experiment (on the 25 April), we transported all individuals back to their pond of origin. Although common frog tadpoles show attraction towards siblings in the late stage of their development [31], non-kin individuals avoid each other at that age [32]. Temporary aggregations of common frog tadpoles can be observed sometimes in natural ponds, but the formation of these collectives is suggested to be the result of mass spawning and/or a response to environmental stimuli [33,34].

During ontogenetic development, same-clutch tadpoles in four containers received control treatment every 2 days, while tadpoles in the fifth container received *Spirulina* treatment. Containers were assigned randomly to these treatments. In the control treatment, we added a 1 : 1 mixture of 60 ml RSW and 60 ml filtered water originating from the container of great pond snails to the tadpoles' water. In the *Spirulina* treatment, we added a 1 : 1 mixture of 60 ml filtered *Spirulina* solution (one organic *Spirulina* tablet [BiOrganik Online Ltd, Hungary] containing 400 mg algae dissolved into 1.5 l RSW) and 60 ml filtered 'snail water' to the 'tadpoles' water. This latter component was used as a conditioned stimulus to indicate the presence of a *Spirulina* food patch for the *Spirulina*-treated individuals during the trials, but provided no such extra information for the control tadpoles. While great pond snails commonly inhabit the same environment as *Rana temporaria* tadpoles and the two species are known to be competitors for the same food sources, tadpoles are competitively dominant over snails and snails facilitate tadpole growth only through an indirect enhancement of tadpoles' preferred food source [35]. We arranged the containers in four rows on the shelves in the laboratory and assigned their positions at random. After 32 days, when all tadpoles surpassed Gosner stage 32 [36] (mortality was approx. 3.1%), we conducted the foraging trials as described below.

## 2.2. Experimental procedure

We randomly allocated tadpoles into C and S collectives, each consisting of eight individuals (one from each clutch). The C collectives contained only control individuals, while in the S collectives, five individuals were control and three individuals were *Spirulina*-treated ($n = 8$–8 replicates, respectively). Tadpoles were deprived of food for 24 h prior to the trials. The arena used during the trial was 60 × 80 × 17 cm in size, its area partitioned with plastic walls (of the same material as the arena itself; electronic supplementary material, figure S1) to form nine compartments (zones) of equal size (the size of the compartments was comparable to that of the rearing containers; electronic supplementary material, figure S1). Between these zones, approximately 4.5 cm wide gates provided passage to the animals. With the inner area of the arena being structured this way, we were able to determine which individuals had a potential influence on which other tadpoles' movement (please note that information on the visual detection range of these tadpoles is lacking).

Video recordings took place on 20 and 21 April 2019. We recorded tadpoles' movement using Panasonic HC V380EP-K (Panasonic Corp., Osaka, Japan) camcorders set to 1920 × 1080 resolution

and 50 frames per second. Four cameras were fixed approximately 125 cm above a horizontal platform onto which we placed four experimental arenas; each camera recorded one arena below. On each of the two recording days, there were three recording rounds, and in each round, we recorded four randomly selected collectives. This resulted in 24 collectives being recorded. We excluded eight out of these 24 collectives from later analysis, because these contained individuals from a failed predator cue treatment. Before recording, we filled each arena with approximately 33 l RSW and then added 330 ml of filtered water from a container of great pond snails; the concentration of this 'snail water' in the arena corresponded to its concentration during the ontogenetic treatments. Approximately 15 min after the administration of the 'snail water', a *Spirulina* tablet was placed in one of the corner zones of each arena (Zone 9; Z9). Then the tadpoles, one at a time and 3 to 10 s apart, were also put into the zone in the opposite corner (Zone 1; Z1), which was temporarily closed with a removable plastic wall. After the removal of these barriers, tadpoles' behaviour in the arena was recorded for 40 min. After each recording round, the arenas were cleaned and prepared for the next round using the following protocol. First, we removed the *Spirulina* tablet and pumped the used water out using an EHEIM compact+ 3000 water pump (EHEIM GmbH & Co. KG, Deizisau, Germany); we also wiped out any *Spriulina* pieces that were left at the bottom. Then, we washed the walls and the bottom of the arenas with 70% ethanol and wiped them clean with a paper towel. Afterwards, the arenas were rinsed with clean water to remove any residual alcohol and the rinse water was poured out as well. This procedure was repeated after each recording round.

In separate activity tests, we also examined if the ontogenetic treatment and the presence of 'snail water' affected individuals' activity as *a priori* expected. For that, we randomly selected one individual from each clutch and from each treatment type and measured their activity in containers identical to their rearing boxes. As we expected large effects of the diet supplement on individual activity [27], we tested our assumptions on a modest number of tadpoles. We placed six containers, each filled with approximately 1.5 l RSW, in the field of view of a camera (four cameras in total). After starting the cameras, we put the animals individually in their randomly arranged containers. After 15 min of recording, 15 ml of 'snail water' was added to each container (in the opposite part of the box to the animal); the obtained concentration corresponded to the concentration of 'snail water' in the ontogenetic treatments. The recordings then continued for another 10 min. These activity tests were conducted on the second day of the recordings, after the completion of the trial videos.

## 2.3. Video analysis and data processing

The video footage was converted to mkv format using FFmpeg (v. N-93894-gecc096513c; [37]) and cropped to size so only the inside area of the arena was visible. By doing so, the original arena identifier was cut off from the footage. We named these new video files in accordance with the day of the recording, the number of recording round and the spatial position of the arena. Video files were then analysed with the event recorder BORIS 7.7.3 [38]. Neither the animals (there were no unique markings) nor the arenas (we cropped the original identifiers) were recognizable on these recordings, so the identity of the animals and collectives remained unknown during the video analysis. We assigned unique BORIS identifiers to the tadpoles based on the order in which individuals left Z1 or based on their spatial coordinates at the start of the trial. The beginning of the recording started when the plastic walls that closed the gates of Z1 were removed and the experimenter left the cameras' field of view; from this time on, 40 min of recording were analysed. We followed the movement of each tadpole in the arena during this period and recorded when the focal tadpole passed from one zone to another (zone transition) as point events and its feedings from the *Spirulina* food patch as state events (i.e. with start time and end time). The obtained data was exported in csv format and then the data on the 16 collectives were merged into a zone transition dataset and a feeding dataset. The former contained 3561 transition events, the latter 23 feeding events. As the order in which we put the tadpoles into the arena was predetermined and documented, original individual identifiers could be assigned to the corresponding BORIS identifiers by tracking the movement of each individual in Z1 between the time of allocation and the start of the trial. To reduce the probability of erroneous identification of differently treated individuals in the S collectives, the three *Spirulina*-treated individuals were always the last to be put into their collectives. From the obtained data, we calculated the time an animal spent in a zone and the number of conspecifics in a zone, we defined the following events and determined the identity of followers and calculated latencies for the following interactions. In seven arenas, some individuals escaped from Z1 before the start of the trial in the C and S collectives, respectively (1–3 tadpoles/arena); these transition events were excluded from later

analysis. In a very few cases, tadpoles transited through the same gate because of being startled or pushed by another individual; these events were not considered as 'following interactions' either.

For the video footage of the activity tests, we used the same procedure as above to avoid observer bias. We discarded the first 5 min from the footages as a short period of acclimatization and analysed the following 10 min (pre-stimulus activity). For the quantification of post-stimulus activity, we used 10 min long recordings after the administration of the 'snail water'. We defined two-state events (with start time and end time): visible (when at least the tail of the animal was clearly visible on the camera) and moving (when the animal was moving its tail). This definition of moving was also applied to compare differences in activity between differently treated tadpoles in previous studies (e.g. [39,40]). Measuring the time when tadpoles were visible was necessary as in some recordings animals that swam close to the wall of the container got out of sight temporarily (time visible (mean ± s.d.): 552.65 ± 92.98 s, $n = 16$). From the obtained data, we calculated the pre- and post-stimulus proportion of time moving (i.e. time moving divided by the time being visible) and used this measure as a proxy for activity in the statistical analysis.

## 2.4. Network construction

We used consecutive zone transitions to define directed and weighted following connections between individuals. A tadpole that left a given zone was considered as a leader for the given movement event, and the first tadpole that followed a leader as a follower. This means that the follower needed to originate from the same zone and arrive in the same adjacent zone as the leader. Instead of imposing a cut-off time for the following events, we used the reciprocal of the time between the transitions of the leader and follower (in s) to estimate the strength of the interaction between the two individuals. Nevertheless, the leader still needed to be in the zone when the follower arrived. Multiple interactions between the same two tadpoles were summed, standardized to time unity (i.e. divided by the Z9 discovery time of the focal individual) and expressed as a 10 min following rate. This standard was chosen arbitrarily to define connections within a realistic time frame; the resulting non-zero tie strengths ranged between 0.0001 and 1.86 (mean ± s.d.: 0.084 ± 0.166). Then, we constructed an individual following network from these connections for each animal. The strength of a directed and weighted connection between any two individuals ($w_{i,j}$) in an individual following network can be expressed by the following equation:

$$w_{i,j} = \frac{\sum(1/\Delta t_{i,j})}{t_i} \times 600,$$

where $i$ is the follower, $j$ is the followed individual, $\Delta t_{i,j}$ is the time elapsed between the two-zone transitions (in seconds) and $t_i$ is the time the follower entered Z9 (in seconds). We also constructed homogeneous networks (i.e. with all possible connections set to 1; [41]) to test whether the diffusion of social information operated independently of the following networks. Furthermore, we calculated the exploration rate for each individual as the number of zone transitions that occurred before the discovery of Z9 and without following another individual, divided by discovery time and expressed as a 10 min rate. This variable was used as a proxy for tadpoles' personal information use during exploration of the arena in the network-based diffusion analysis.

## 2.5. Statistical analysis

We conducted all statistical analyses in R v. 3.6.1 [42]. We fitted a linear mixed-effect model (LMM) to examine how *Spirulina* treatment, the presence of 'snail water' and their interaction affected the proportion of time moving, a proxy for activity, in the activity test. We included 'individual identity' as a random variable in this model. We fitted a LMM to investigate the difference between the C and S collectives in the number of individual zone transitions and a binomial generalized linear mixed-effect model (GLMM) to investigate the effect of collective type on the proportion of zone transitions that elicited a following. In the former model, the response variable was square-root transformed to improve its fit to normal distribution. We used binomial GLMM with a Bernoulli response to examine the difference in the probability of feeding from the food patch between collective types. All LMMs and GLMMs were fitted using the 'lme4' R package [43]. LMMs were fitted with restricted maximum likelihood estimation and Satterthwaite's approximation for degrees of freedom. We fitted a mixed-effects Cox model to examine if the probability of leaving Z1 and the probability of reaching Z9 differed between collective types using the 'coxme' R package [44]. We included 'collective identity' as a random variable into all these latter mixed-effects models. Requirements of the fitted models were checked by plot diagnosis and by testing for overdispersion (where relevant). Compliance with the proportional

hazard assumption was confirmed in the fitted Cox models. In the case of GLMMs, we used the 'DHARMa' R package [45] for residual diagnostics. For all predictor levels in the fitted models (electronic supplementary material table S1), we reported estimated marginal means (EMMs) with 95% CIs and corresponding contrasts as unstandardized effects on the response scale; these were calculated using the relevant functions of the 'emmeans' R package [46]. All tests were two-tailed with α set to 0.05.

We used the time-of-acquisition (continuous TADA) variant of NBDA [27,31,32], in which models are fitted to time of acquisition data with a social transmission component with the rate of transmission between informed and naive animals being proportional to their connection in the network. In the extended time of acquisition variant of the NBDA model [47], individual-level variables that may influence the rate of acquisition of a novel trait can be incorporated. If the asocial and social effects interact additively, then the rate at which individual $i$ solves a task or discovers a food patch at time $t$ is given by

$$\lambda_i(t) = \lambda_0(1 - z_i(t))\left(s\sum_{j=1}^{N}(a_{i,j}\,z_j(t)) + (1-s)\exp\left(\sum_{k=1}^{V}\beta_{k,}x_{k,i}\right)\right),$$

or individual-level variables can be incorporated using a multiplicative model

$$\lambda_i(t) = \lambda_0(1 - z_i(t))\left(s\sum_{j=1}^{N}(a_{i,j}\,z_j(t)) + (1-s)\right)\exp\left(\sum_{k=1}^{V}\beta_{k,}x_{k,i}\right),$$

where $\lambda_0$ is the baseline rate of acquisition, $z_i(t)$ is the status of individual $i$ at time $t$, $s \geq 0$ is a parameter determining the rate of social transmission between individuals per unit of network connection, $a_{i,j}$ is the network connection between individual $i$ and $j$, $z_j(t)$ is the status of $j$ at time $t$ (1 indicates informed and 0 indicates naive), $N$ is the number of individuals, $\beta_k$ is the coefficient determining the effect of variable $k$, $x_{k,i}$ is the value of variable $k$ for individual $i$, and $V$ is the number of individual-level variables in the model [47,48]. We extended the available script (v. 1.2.13; https://lalandlab.st-andrews.ac.uk/freeware/) in a way that it takes individual networks as input data, and calculates social transmission metric for each animal from its individual network. We fitted models with the baseline rate of acquisition being either constant or following a gamma distribution, and with the social and asocial variables (if both present) interacting either additively or multiplicatively. We used the following settings for social transmission ('sParam' in the NBDA context): it was present either only in the C collectives or only in the S collectives, or it was the same in both collective types, or different in the two collective types, or absent (asocial models). We tested the effect of exploration rate on the discovery of Z9 as an asocial individual-level variable. We also fitted models using homogeneous networks in which all individuals had the same influence over each other; $w_{ij} = 1$ for all $i$ and $j$ [41]; by doing so, we tested whether the diffusion of social information takes place independently of the following networks.

In total, we fitted 52 models (see in the electronic supplementary material) and ranked them according to their predictive power using Akaike information criteria corrected for small sample sizes (AICc) and corresponding Akaike weights [49,50] (electronic supplementary material, table S2). We used model averaging to obtain estimates for the investigated model parameters [50]. Because of the highly asymmetrical profile likelihoods of the social transmission parameters in our models, we did not compute unconditional standard errors; instead, we derived 95% confidence intervals using profile likelihood techniques [51]. In the case of two models (see in electronic supplementary material, table S2), the default estimation procedure produced NaNs during the profile likelihood CI calculations (for $s_S$ and for exploration rate in the one-one model, respectively), so we verified the values of the lower and upper bounds by using the 'L-BFGS-B' optimization method [52]. We based our inference on the model-averaged parameter estimates calculated from model parameters in the 'best models' set (within 5 ΔAICc with the best-fitting model) and calculated conditional 95% profile likelihood confidence intervals for these estimates from the best predictive model that included a given parameter (as in [53]).

We used the estimated social transmission parameter in the C collectives ($s_C$) to calculate the proportion of those discovery events that occurred by social transmission (as opposed to asocial learning). For each individual $i$ that discovered Z9 at time $t_i$, we calculated the probability of social transmission as

$$p_{\text{social,i}} = \frac{s_c \sum_j a_{i,j}(t_i)z_j(t_i)}{1 + s_c \sum_j a_{i,j}(t_i)z_j(t_i)}$$

then computed the mean of $p_{\text{social,i}}$ across tadpoles [15].

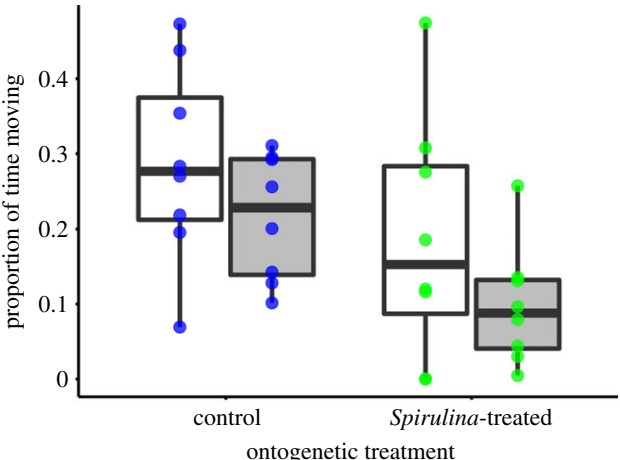

**Figure 1.** The effect of the ontogenetic treatment and the administration of 'snail water' to the proportion of time moving, a proxy for activity, in the activity test. Boxplots show the median and interquartile range; $n = 8$ tadpoles for each treatment type. Whiskers show values within 1.5-fold of the interquartile range, dots indicate individual values. White boxplots indicate pre-stimulus, while grey boxplots denote post-stimulus responses in tadpoles belonging to one of the two ontogenetic treatments.

## 3. Results

In the activity tests, we found that the proportion of time moving was significantly lower in the *Spirulina*-treated tadpoles compared to the control individuals (EMMs with 95% CI: 0.14 [0.07, 0.21] versus 0.25 [0.18, 0.32]; C–S contrast ± s.e.: 0.11 ± 0.05, $t$-ratio$_{20.9}$ = 2.31, $p = 0.031$; figure 1). The administration of the associative cue, contrary to our expectations, had a negative effect on the activity of both *Spirulina*-treated and control individuals (0.16 [0.1, 0.22] versus 0.24 [0.18, 0.30]; post–pre contrast ± s.e.: −0.08 ± 0.03, $t$-ratio$_{20.9}$ = −2.31, $p = 0.032$; table 1). Thus, 'snail water' did not compensate for the effect of *Spirulina* supplementation in tutor tadpoles, implying that *Spirulina*-treated individuals did not associate 'snail water' cues with the presence of *Spirulina* and thus did not increased their food searching activity when 'snail water' was added to their water.

During the foraging trials, we recorded 3561 zone transition events and 23 feeding events. Ninety-three out of 128 tadpoles (72.66%) entered Z9 containing the food patch, while 12 individuals (9.38%) fed from the patch, two tadpoles from one of the C collectives and ten individuals (five control and five *Spirulina*-treated individuals) from five S collectives. We found that the probability of leaving Z1 was significantly higher in the C collectives than in the S collectives (marginal responses with 95% CI: 1.34 [1.09, 1.64] versus 0.75 [0.61, 0.92]; C/S ratio ± s.e.: 1.79 ± 0.37, $z$-ratio = 2.81, $p = 0.005$; table 1). The number of individual zone transitions did not differ between the two collective types (EMMs with 95% CI: 26.5 [17.5, 35.5] versus 17.5 [10.2, 24.8]; C–S contrast ± s.e.: 9.01 ± 5.4, $t$-ratio$_{14}$ = 1.67, $p = 0.117$; table 1). However, the proportion of those zone transitions that elicited a following was significantly higher in the C collectives than in the S collectives (marginal probabilities with 95% CI: 0.27 [0.24, 0.31] versus 0.22 [0.19, 0.25]; odds ratio ± s.e.: 1.32 ± 0.16, $z$-ratio = 2.28, $p = 0.022$; table 1). The probability of reaching Z9 was also significantly higher in the C collectives compared to the S collectives (marginal responses with 95% CI: 1.26 [1.02, 1.55] versus 0.8 [0.65, 0.98]; C/S ratio ± s.e.: 1.58 ± 0.33, $z$-ratio = 2.18, $p = 0.029$; table 1). The probability of feeding was nevertheless marginally lower in the C collectives than in the S collectives (marginal probabilities with 95% CI: 0.02 [0.002, 0.14] versus 0.12 [0.04, 0.31]; odds ratio ± s.e.: 0.14 ± 0.15, $z$-ratio = −1.85, $p = 0.064$; table 1).

In the NBDA, models fitted with following-based individual networks had a substantially higher overall support (97.89%; electronic supplementary material, table S1) compared to those fitted with homogeneous networks (1.55%) or without a social parameter (i.e. asocial models; 0.56%). In the 'best models' set (i.e. those nine models within 5 ΔAICc with the best-fitting model; 93.99% overall support), social transmission parameter in the C collectives was estimated to be higher than zero in all models, but it was either constrained to or not different from zero in the S collectives (electronic supplementary material, table S2). This result indicates that following those conspecifics that already reached Z9 during zone transitions facilitated the discovery of Z9 by naive tadpoles in the C collectives, but that effect was negligible in the S collectives. Corresponding to the model-averaged value of the social transmission parameter (table 2), we estimated that 21.52% [9.34, 27.51%] of the

**Table 1.** Test statistics and the significance of the investigated explanatory variables from the fitted models. Significant predictors are shown in italics; random terms are given as s.d. $\pm$ 95% profile confidence intervals. To estimate the significance of potential predictors in the fitted models, we applied type III Wald $\chi^2$-tests using the Anova function of the 'car' R package [54]. Test statistic and $p$-value for a non-significant predictor was obtained by including it into the final model.

| response variable | model type | random term | predictors | $\chi^2$ | d.f. | $p$ |
|---|---|---|---|---|---|---|
| proportion of time moving | LMM | 'Individual': 0.07 [0, 0.12] | intercept | 30.99 | 1 | <0.001 |
| | | | *treatment type* | *5.33* | *1* | *0.021* |
| | | | *presence of 'snail water'* | *5.31* | *1* | *0.021* |
| | | | treatment type × presence of 'snail water' | 0.05 | 1 | 0.825 |
| time to leave Z1 | mixed-effects Cox model | 'Collective': 0.15 [0, 0.5] | intercept | — | — | — |
| | | | *collective type* | *7.91* | *1* | *0.005* |
| zone transitions (sqrt-transformed) | LMM | 'Collective': 0.8 [0, 1.36] | intercept | 160.26 | 1 | <0.001 |
| | | | collective type | 2.81 | 1 | 0.093 |
| proportion of zone transitions that elicited a following | binomial GLMM | 'Collective': 0.18 [0.07, 0.32] | intercept | 142.64 | 1 | <0.001 |
| | | | *collective type* | *5.22* | *1* | *0.022* |
| time to reach Z9 | mixed-effects Cox model | 'Collective': 0.02 [0, 0.46] | intercept | — | — | — |
| | | | *collective type* | *4.75* | *1* | *0.029* |
| probability of feeding | binomial GLMM | 'Collective': 1.01 [0, 3.22] | intercept | 13.71 | 1 | <0.001 |
| | | | collective type | 3.42 | 1 | 0.064 |

**Table 2.** Estimated rates of social transmission ($s$) in the following networks based on the models from the 'best models' set. $s_C$ denotes the social transmission parameter in the C collectives, whereas $s_S$ is the social transmission parameter in the S collectives.

| model parameters | $\sum$Akaike weights | model-averaged estimates with 95% CI[a] |
|---|---|---|
| $s_C$ | 0.81 | 7.68 [1.77, 14.01] |
| $s_S$ | 0.21 | 1.03 [−1.66, 7.09] |
| the same $s$ in both collectives | 0.14 | 4.73 [0.90, 10.10] |
| exploration rate | 0.94 | 0.07 [0.06, 0.09] |

[a]Confidence intervals of the model-averaged estimates were obtained using profile likelihood techniques from the highest ranked model that included a given parameter (as in [53]).

observed discovery events occurred through social diffusion in the C collectives. Exploration rate (measured as the rate of zone transitions without following another conspecific before the discovery of Z9) also affected the time of discovery in all models in the 'best models' set (table 2), implying that an increased use of personal information also resulted in increased discovery rate (figure 2).

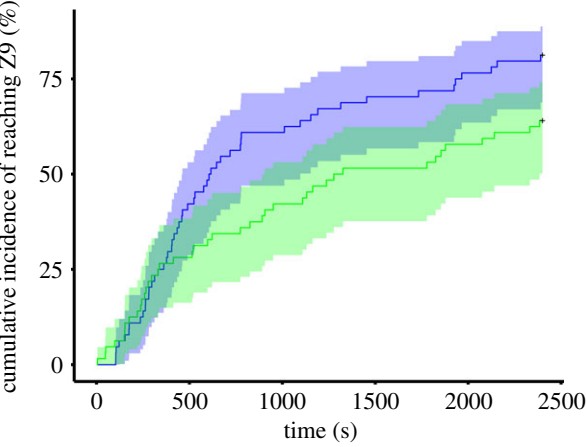

**Figure 2.** Kaplan–Meier curves for the cumulative incidences of reaching Z9 in the two collective types (blue: C collectives; green: S collectives). Curves are shown with 95% confidence intervals. For graphical presentation only, we used the *survfit* function of the 'survival' R package [55] without including the random term.

## 4. Discussion

In this study, we investigated how common frog tadpoles used social cues during a foraging task. We found that animals exploited simple social cues provided by conspecifics: the movement of tadpoles affected the spatial decisions of others in the collective during the exploration of their environment and following those conspecifics that already discovered the presented food patch contributed to the collective-level discovery of that patch. However, this social effect was detectable only in those collectives where tadpoles had a similar activity level, but not in those where less active tutors, i.e. *Spirulina*-treated tadpoles, were also present. Because of this social effect, the probability of finding the food patch was higher in the control collectives, but the probability of feeding was still marginally lower than in the tutored collectives. This was most probably due to a lasting neophobia toward the novel food type in the control individuals. Thus, tutors failed to facilitate fast resource exploitation in their collectives because their low activity contributed to movement patterns (in terms of leaving of the start zone, proportion of followings and reaching the target zone) that hindered the discovery of the presented food patch. To our knowledge, this study is the first to quantify how temporary following incidences induce behavioural correlations among non-grouping individuals and affect resource discovery in their collectives. The influence of one another's movements on discovery rates in a foraging context has previously been observed in cohesive groups of three-spine sticklebacks (*Gasterosteous aculeatus*) and termed as 'untransmitted social effect' [56,57]. We propose that the same phenomenon is likely to underlie the observed social diffusion in the control tadpole collectives as well, indicating that the routine behaviour of conspecifics may be an important source of social information in temporary feeding aggregations even in the absence of social attraction between individuals.

One may argue that no social information has been shared in this experimental set-up, and our findings are simply a function of tadpole density introduced into the arena. We agree that the density of tadpoles is likely to affect the opportunity rate to observe others (or simply the chance to move in the same direction as another conspecific), but that would increase only the proportion of followings between tadpoles during zone transitions, and not necessarily the effect of following connections on the rate of patch discovery. The applied density was used to avoid any repulsion effects that could have biased our estimation of potential social effects (such repulsion was observed between non-kin common frog tadpoles in a previous study; [32]) and was the same density at which tadpoles were raised during their ontogenetic development. In group-living species, individuals exhibit social attraction toward conspecifics and thus maintain spatial proximity with group-mates [58], but non-grouping individuals do not necessarily deviate from their own preferred behaviours to maintain cohesion [59,60]. Thus, our results should not be viewed as the result of intrinsic movement coordination either. With time, all tadpoles would have discovered the food patch (as individual exploration also affected patch discovery rate), and after overcoming initial neophobia and without other available food source, fed from the *Spirulina* patch. The main question we addressed here was whether or not tutors can facilitate this process (discovery and exploitation of the presented food patch) within the studied time frame. Contrary to our prediction, findings indicate that phenotypic

differences between individuals rather counteracted the emergence of such a facilitation effect in the studied tadpole collectives.

We found that *Spirulina*-treated individuals were less active than naive individuals in a separate activity test, which is in line with our expectation as most anuran tadpoles raised in ad libitum condition are known to be less active than food-restricted conspecifics (in common frog tadpoles: e.g. [61]). In our study, the applied ontogenetic *Spirulina* treatment most likely provided a surplus of resource to the treated individuals, which resulted in their generally reduced movement activity compared to the control tadpoles that received low-quality food throughout the experiment. Contrary to our expectation, however, *Spirulina*-treated tadpoles did not associate 'snail water' with the presence of food and did not increase their activity to search for food when only this cue was added to the water. We believe that this lack of increase in activity, in turn, prevented them from being effective tutors in food patch exploitation in their collectives. This does not necessarily mean, however, that social information use was not present in the tutored collectives. Reduced activity can be a social cue for indicating predation threat in tadpoles [39,40,62], so it is possible that the behaviour of tutors did not provide adaptive information to conspecific observers in a foraging context, but their low activity level induced similar behavioural adjustments in others leading to the observed differences between the studied collective types. Alternatively, reduced activity of satiated conspecifics failed to decrease the time required for naive tadpoles to discover and feed from the presented food patch because the relative difference in activity did not make tutors conspicuous enough to others and thus were not recognized as social cue producers [27]. Conditions under which social information is more beneficial than personally gained information can be context-specific and could depend on the presence of additional environmental cues as well. For instance, European minnows (*Phoxinus phoxinus*) changed their resource patch preference after observing demonstrators only under high-risk conditions, but rather relied on personal information in the absence of threat [63]. On the contrary, a foraging task solution was socially transmitted at a higher rate in Trinidadian guppy (*Poecilia reticulata*) shoals when the background predation risk was low [22]. Nevertheless, our results imply that the advantage of gaining information regarding the location and exploitability of unevenly distributed food resources from more experienced but less active conspecifics can be limited in temporary aggregations of non-grouping animals.

Although we did not find strong evidence for the beneficial influence of tutor individuals on conspecifics' feeding performance, our findings indicate that there were correlations between individual activity levels, discovery rates of a food patch (also by independent exploration) and the presence of social facilitation in patch discovery in the studied tadpole collectives. While within-group heterogeneity is known to promote the emergence of leadership, group coordination and the spread of innovations in social groups [6,12,64,65], our work also illustrates that phenotypic heterogeneity in less stable groups, including temporary aggregations, does not necessarily lead to functional benefits for groups or their members.

Ethics. All sampling procedures and experimental manipulations reported in this study were approved by the national authority of the Middle-Danube-Valley Inspectorate for Environmental Protection, Nature Conservation and Water Management, Hungary, who issued the permission to capture and conduct an experiment on the animals (PE/EA/1540-7/2018 and PE-06/KTF/32929-8/2018, PE-06/KTF/32929-9/2018, PE-06/KTF/32929-10/2018, PE-06/KTF/32929-11/2018). This work also complies with all applicable institutional and national guidelines for the care and use of animals in research.

Data accessibility. Custom R codes and all data used in the statistical analyses are available from Figshare (doi:10.6084/m9.figshare.11854845).

Authors' contributions. Z.T. conceived and designed the study, performed the experiment, analysed the data, wrote the manuscript, and revised and edited the manuscript. B.J. participated in the video analysis and contributed to the text and revisions.

Competing interests. The authors have no conflict of interest to declare.

Funding. Z.T. was financially supported by the Prémium Postdoctoral Research Programme of the Hungarian Academy of Sciences (grant no. PREMIUM-2018-198) and B.J. by the Young Researcher Programme of the Hungarian Academy of Sciences (grant no. Mv-33/2019).

Acknowledgements. We thank Gergely Tarján for his help in the laboratory and in the video analysis. We are also indebted to the Pilisi Parkerdő Zrt. for allowing us to use their forestry roads during the collection of the animals.

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
