## [Peer Review File · Royal Society Open Science]

Review History

RSOS-200876.R0 (Original submission)

Review form: Reviewer 1

Is the manuscript scientifically sound in its present form?

No

Are the interpretations and conclusions justified by the results?

Yes

Is the language acceptable?

Yes

Do you have any ethical concerns with this paper?

No

Have you any concerns about statistical analyses in this paper?

Yes

Recommendation?

Reject

Comments to the Author(s)

This is an interesting manuscript that aims to investigate whether social information spreads in temporary aggregations of tadpoles. The authors make valid points about the importance of the network structure in influencing trends about social learning/information transmission.

However, I did not feel that the methods fulfilled the research question. I have three primary suggestions below that I believe could improve this paper.

1. Small sample-size

For an experiment, I am surprised authors have considered only 16 individuals, 8 in a control group and other 8 with the Spirulina treatment. This reduces the power of study and increases the margin of error, mostly leading to Type 2 errors. I am aware it is frustrating to hear this after the experiments have been conducted, but this number is extremely small and if possible, I highly recommend authors to repeat the experiments with higher sample size, more replicas for the control and treatment groups.

2. Social information transmission

Authors consider the premise that social cue has been transmitted based on individuals' spatial decisions rather than on the nutrition per se of the spirulina. This seems to be a rather stretch of the question under study since the social cue under interest has not been transmitted in most of the cases. Further investigation should be conducted to address this question.

3. The effect of Spirulina on treated individuals

Authors defend in the discussion section (e.g., lines 296-299) that "individuals remained satiated and did not start exploring their environment when the associative cue was added to their water" to explain the lack of social transmission on the experiment group. This seems to be a rather baseline information to determine the timing of the experiment and the food used. Another question is: if Spirulina causes a decrease in the activity of the individuals, couldn't the authors use another food?

Review form: Reviewer 2

Is the manuscript scientifically sound in its present form?

No

Are the interpretations and conclusions justified by the results?

Yes

Is the language acceptable?

Yes

Do you have any ethical concerns with this paper?

No

Have you any concerns about statistical analyses in this paper?

No

Recommendation?

Major revision is needed (please make suggestions in comments)

Comments to the Author(s)

In this study the authors collected data on leader-follower interactions in experimental groups of tadpoles. The tadpoles were moving through an arena that was divided into separate compartments, one of which contained food. The authors fitted models to their network data to understand what best predicted how tadpoles found the food. They showed that naïve tadpoles that were following others that had already found the food once were more likely to encounter the food themselves, demonstrating an untransmitted social effect shaping their foraging behaviour. The authors tested two sets of treatment groups of tadpoles, half of which contained some members that had previously been exposed to snail-conditioned water paired with spirulina (the food in the test itself) and half which only contained members exposed to just the snail-conditioned water. It was predicted that the groups containing the snail and spirulina conditioned members might find the food faster (a result of classical conditioning?), but in fact it was the snail-only groups that found the food faster.

Overall I found this an interesting study and the leader-follower effect on food discovery is clear. The analyses are appropriate for studying network-based diffusion. The paper will appeal to researchers who are interested in collective behaviour and movement and social foraging.

There is some scope for improving the presentation of the paper. Some of the terminology is confusing but this can easily be addressed with a revision. In particular, I was confused by the use of term 'knowledgeable' in the title and elsewhere. I took this at first to refer to the 'snail-conditioned water paired with spirulina' treatment groups, but in fact it refers to the tadpoles that have already visited the food once, irrespective of treatment group. This should be made clear early on. The snail + spirulina treatment confused me a bit- was this intended to lead to classical conditioning, so that conditioned tadpoles searched more actively at test? If so, did it work? Do you have the data to show this?

Some other comments

L42: Please define 'differentiated social structure'

L42: Non-social is probably the wrong term since even animals that live alone are exposed to social cues and interactions. Non-grouping is perhaps a better term.

L44: What is meant by 'temporal aggregations'?

L49: Connections need not be random?

L92: Can you be sure if the tadpoles were non-kin if they were spawned in the wild? Could multiple egg clutches be fertilised by the same male?

L96: Perhaps worth mentioning that the snail water was used as a conditioned stimulus first (I presume it was?), otherwise readers will be wondering why you added them. Can you provide more detail on the responses of tadpoles to this CS- do they respond, and if so, how? This needs to be demonstrated

L124 & S1: Why the partitions? Why not use an open arena? Could the partitions have impeded movement? It certainly seems like they could have prevented the transmission of visual information between tadpoles.

L142-144: More information on how the leader-follower interaction was determined is needed here. Did you impose a cut off time, so that a follower event could only occur within a fixed time?

Did the follower need to originate from the same compartment as the leader? Did the leader still need to be in the new compartment when the follower arrived?

ESM: I think that all of the material in the ESM (except perhaps the tables) should be moved to the main text- this info is all important for understanding the procedures and analyses, and since RSOS has no word limit (as far as I'm aware anyway), there is no reason to separate the material between two different documents.

Decision letter (RSOS-200876.R0)

Dear Dr Tóth,

Manuscript ID RSOS-200876 entitled "Knowledgeable individuals do not facilitate rapid resource exploitation in tadpole aggregations" which you submitted to Royal Society Open Science, has been reviewed. The comments from reviewers are included at the bottom of this letter.

In view of the criticisms of the reviewers, the manuscript has been rejected in its current form. However, a new manuscript may be submitted which takes into consideration these comments.

Please note that resubmitting your manuscript does not guarantee eventual acceptance, and that your resubmission will be subject to peer review before a decision is made.

Your resubmitted manuscript should be submitted by 27-Jan-2021. If you are unable to submit by this date please contact the Editorial Office.

on behalf of Dr Sean Rands (Associate Editor) and Pete Smith (Subject Editor)
openscience@royalsociety.org

Associate Editor Comments to Author (Dr Sean Rands):

Firstly, please accept my apologies for the delay in getting a decision to you, as I've been a little delayed by the ongoing epidemic. I hope you are safe and well.

Two reviewers have commented on the manuscript, and their reports should be attached. Reviewer #1 raises concerns about sample size (I appreciate that these numbers refer to the

different collectives treated rather than individuals), and has some questions about the assumptions behind the experiments. Reviewer #2 raises some more general concerns about specific details.

Based on their reports and my own reading, I share the concerns about sample size, and would like to be reassured that there is enough power in what you did for the results to be meaningful. I suspect some post hoc power analysis would help to justify your result, if it confirms that there is enough power. I'm therefore going to recommend that the manuscript is rejected in its current form, but that you are encouraged to resubmit a revised version if you can justify the sample size. At the same time, please address the rest of the reviewers' comments (noting that I agree with reviewer #2's suggestion that methodology in the supplementary material should be moved into the main text, although I'd keep the AIC tables where they are).

Reviewers' Comments to Author:

Reviewer: 1

Comments to the Author(s)

This is an interesting manuscript that aims to investigate whether social information spreads in temporary aggregations of tadpoles. The authors make valid points about the importance of the network structure in influencing trends about social learning/information transmission. However, I did not feel that the methods fulfilled the research question. I have three primary suggestions below that I believe could improve this paper.

1. Small sample-size

For an experiment, I am surprised authors have considered only 16 individuals, 8 in a control group and other 8 eight with the Spirulina treatment. This reduces the power of study and increases the margin of error, mostly leading to Type 2 errors. I am aware it is frustrating to hear this after the experiments have been conducted, but this number is extremely small and if possible, I highly recommend authors to repeat the experiments with higher sample size, more replicas for the control and treatment groups.

2. Social information transmission

Authors consider the premise that social cue has been transmitted based on individuals' spatial decisions rather than on the nutrition per si of the spirulina. This seems to be a rather stretch of the question under study since the social cue under interest has not been transmitted in most of the cases. Further investigation should be conducted to address this question.

3. The effect of Spirulina on treated individuals

Authors defend in the discussion section (e.g., lines 296-299) that "individuals remained satiated and did not start exploring their environment when the associative cue was added to their water" to explain the lack of social transmission on the experiment group. This seems to be a rather baseline information to determine the timing of the experiment and the food used. Another question is: if Spirulina causes a decrease in the activity of the individuals, couldn't the authors use another food?

Reviewer: 2

Comments to the Author(s)

In this study the authors collected data on leader-follower interactions in experimental groups of tadpoles. The tadpoles were moving though an arena that was divided into separate compartments, one of which contained food. The authors fitted models to their network data to understand what best predicted how tadpoles found the food. They showed that naïve tadpoles

that were following others that had already found the food once were more likely to encounter the food themselves, demonstrating an untransmitted social effect shaping their foraging behaviour. The authors tested two sets treatment groups of tadpoles, half of which contained some members that had previously exposed to snail-conditioned water paired with spirulina (the food in the test itself) and half which only contained members exposed to just the snail-conditioned water. It was predicted that the groups containing the snail and spirulina conditioned members might find the food faster (a result of classical conditioning?), but in fact it was the snail-only groups that found the food faster.

Overall I found this an interesting study and the leader-follower effect on food discovery is clear. The analyses are appropriate for studying network-based diffusion. The paper will appeal to researchers who are interested in collective behaviour and movement and social foraging.

There is some scope for improving the presentation of the paper. Some of the terminology is confusing but this can easily be addressed with a revision. In particular, I was confused by the use of term 'knowledgeable' in the title and elsewhere. I took this at first to refer to the 'snail-conditioned water paired with spirulina' treatment groups, but in fact it refers to the tadpoles that have already visited the food once, irrespective of treatment group. This should be made clear early on. The snail + spirulina treatment confused me a bit- was this intended to lead to classical conditioning, so that conditioned tadpoles searched more actively at test? If so, did it work? Do you have the data to show this?

Some other comments

L42: Please define 'differentiated social structure'

L42: Non-social is probably the wrong term since even animals that live alone are exposed to social cues and interactions. Non-grouping is perhaps a better term.

L44: What is meant by 'temporal aggregations'?

L49: Connections need not be random?

L92: Can you be sure if the tadpoles were non-kin if they were spawned in the wild? Could multiple egg clutches be fertilised by the same male?

L96: Perhaps worth mentioning that the snail water was used as a conditioned stimulus first (I presume it was?), otherwise readers will be wondering why you added them. Can you provide more detail on the responses of tadpoles to this CS- do they respond, and if so, how? This needs to be demonstrated

L124 & S1: Why the partitions? Why not use an open arena? Could the partitions have impeded movement? It certainly seems like they could have prevented the transmission of visual information between tadpoles.

L142-144: More information on how the leader-follower interaction was determined is needed here. Did you impose a cut off time, so that a follower event could only occur within a fixed time? Did the follower need to originate from the same compartment as the leader? Did the leader still need to be in the new compartment when the follower arrived?

ESM: I think that all of the material in the ESM (except perhaps the tables) should be moved to the main text- this info is all important for understanding the procedures and analyses, and since

RSOS has no word limit (as far as I'm aware anyway), there is no reason to separate the material between two different documents.

Author's Response to Decision Letter for (RSOS-200876.R0)

See Appendix A.

RSOS-202288.R0

Review form: Reviewer 2

Is the manuscript scientifically sound in its present form?

Yes

Are the interpretations and conclusions justified by the results?

Yes

Is the language acceptable?

Yes

Do you have any ethical concerns with this paper?

No

Have you any concerns about statistical analyses in this paper?

No

Recommendation?

Accept as is

Comments to the Author(s)

In this paper the authors demonstrate that among aggregations of non-grouping tadpoles (which are non-socially attracted to one another but which nevertheless may occur in close proximity to one another), individuals respond to social information in the form of movement cues, allowing them to locate patches of food. The authors asked whether experienced 'tutor' tadpoles facilitated food discovery more effectively or rapidly compared to aggregations where tutors were not present. Instead they saw the opposite effect- socially facilitated food discovery occurred in aggregations without tutors, which may result from differences in the movement cues produced by experienced versus inexperienced tadpoles. Recently a number of studies have appeared that highlight the importance of social information in influencing behaviour in non-grouping species, so this paper is timely. The authors have used network-based diffusion analysis to estimate social influences on behaviour. To the best of my knowledge this has only been applied to group-living animals previously, and it interesting to see it applied to a no-grouping species here.

I previously reviewed a version of this manuscript for RSOS. At the time I thought that the paper was sound but suggested that a number of corrections and clarifications be made. The authors have addressed these concerns, specifically regarding terminology. Bringing the information on

analyses into the main text from the ESM has made the paper much easier to follow. I have no further comments and recommend that the manuscript be accepted.

Decision letter (RSOS-202288.R0)

Dear Dr Tóth,

I am pleased to inform you that your manuscript entitled "Tutors do not facilitate rapid resource exploitation in temporary tadpole aggregations" is now accepted for publication in Royal Society Open Science.

on behalf of Dr Sean Rands (Associate Editor) and Pete Smith (Subject Editor)
openscience@royalsociety.org

Subject Editor Comments to Author (Pete Smith)

The major concerns have been addressed. Sample size was one of the main concerns in the previous round of revision - the authors have provided a thorough response and justified this within their point-by-point response letter. Thanks for your careful revision.

Reviewer comments to Author:

Reviewer: 2

Comments to the Author(s)

In this paper the authors demonstrate that among aggregations of non-grouping tadpoles (which are non-socially attracted to one another but which nevertheless may occur in close proximity to one another), individuals respond to social information in the form of movement cues, allowing them to locate patches of food. The authors asked whether experienced 'tutor' tadpoles facilitated food discovery more effectively or rapidly compared to aggregations where tutors were not present. Instead they saw the opposite effect- socially facilitated food discovery occurred in aggregations without tutors, which may result from differences in the movement cues produced by experienced versus inexperienced tadpoles. Recently a number of studies have appeared that highlight the importance of social information in influencing behaviour in non-grouping species, so this paper is timely. The authors have used network-based diffusion analysis to estimate social influences on behaviour. To the best of my knowledge this has only been applied to group-living animals previously, and it interesting to see it applied to a no-grouping species here.

I previously reviewed a version of this manuscript for RSOS. At the time I thought that the paper was sound but suggested that a number of corrections and clarifications be made. The authors have addressed these concerns, specifically regarding terminology. Bringing the information on analyses into the main text from the ESM has made the paper much easier to follow. I have no further comments and recommend that the manuscript be accepted.

Appendix A

Dear Prof. Pete Smith FRS,

We have revised the manuscript entitled "Knowledgeable individuals do not facilitate rapid resource exploitation in tadpole aggregations" (manuscript ID: RSOS-200876) following the recommendations of the Associate Editor and the two Reviewers. We are grateful for the helpful suggestions and comments that aimed to improve this manuscript. In the revised ms, we replaced all group-level statistical analyses with mixed effects models fitted on individual-level data to emphasize that our main findings were based on the behavioural analysis of 128 individuals. We included more details about the administered conditional stimulus as well and put more emphasis on discussing the findings of the activity tests to highlight this important aspect of the experimental design. We also moved the relevant methodological descriptions from the ESM to the main text to improve clarity as requested by the Associate Editor and Reviewer 2. Furthermore, we applied several changes to make the text more readable throughout the ms. We have addressed all comments of the Associate Editor and the Reviewers, and we hope that you will find this revised version acceptable for publication in Royal Society Open Science.

We provided a detailed list of our responses to the comments of the Reviewers. For clarity, after quoting each comment, our response is written in bold. The changes in the text can be tracked in the "TrackChanges" version of the revised manuscript.

Thank you for your consideration,

Zoltán Tóth,
corresponding author

Responses to the Associate Editor's Comments

1.1. Firstly, please accept my apologies for the delay in getting a decision to you, as I've been a little delayed by the ongoing epidemic. I hope you are safe and well.

We thank the opportunity provided to improve our manuscript. We are especially grateful for the extended deadline for the resubmission; it proved to be extremely valuable in the end. These are difficult times as our lives are profoundly affected by the COVID-19 pandemic, so please do not worry about a little delay. I hope all is well with you, too.

1.2. Two reviewers have commented on the manuscript, and their reports should be attached. Reviewer #1 raises concerns about sample size (I appreciate that these numbers refer to the different collectives treated rather than individuals), and has some questions about the assumptions behind the experiments. Reviewer #2 raises some more general concerns about specific details. Based on their reports and my own reading, I share the concerns about sample size, and would like to be reassured that there is enough power in what you did for the results to be meaningful. I suspect some post hoc power analysis would help to justify your result, if it confirms that there is enough power. I'm therefore going to recommend that the manuscript is rejected in its current form, but that you are encouraged to resubmit a revised version if you can justify the sample size.

The main experiment of the study was conducted on 128 animals that were divided into eight tutored and eight untutored collectives. We propose that having a comparable number of replicates and individuals in total is not unprecedented in similar studies (Liker

& Bókonyi 2009, Farine et al. 2015, Hasenjager et al. 2020, Canteloup et al. 2020). Nevertheless, we agree with the Editor that this sample size limited our ability to examine individual-level effects within collectives or detect collective-level effects with low effect sizes, and therefore we carefully rephrased the text when interpreting non-significant results. We also improved the statistical analysis of all response variables by applying mixed effects models fitted on individual-level data; into these models, we incorporated 'collective identity' as a random term. Mixed models are vastly superior in controlling for Type I errors than alternative approaches and consequently results from mixed models are more likely to generalize to new observations (e.g., Luke 2016).

Using post hoc power analysis for the justification of statistical results has been heavily criticized for being logically flawed, however. Briefly, non-significant effects will always have low observed power because the calculated power is a monotone function of the *P*-value and therefore contains no additional helpful information; please see the references in this topic listed below. Instead, these works have advocated reporting confidence intervals and/or appropriate effect sizes. In the case of mixed effects models, due to the way that variance is partitioned there does not exist an agreed upon way to calculate standard effect sizes for individual model terms such as main effects or interactions. Because of that, we reported unstandardized effect sizes (in line with the arguments in Kline 2004, Nakagawa & Cuthill 2007), i.e., effects on the response scale, in the revised manuscript along with estimated marginal means/probabilities and 95% confidence intervals for each predictor level (lines 286-289, 347-374).

References related to post hoc power analysis:

- Colegrave, N., & Ruxton, G. D. (2003). Confidence intervals are a more useful complement to nonsignificant tests than are power calculations. *Behavioral Ecology*, 14(3), 446-447.
- Nakagawa, S., & Foster, T. M. (2004). The case against retrospective statistical power analyses with an introduction to power analysis. *Acta Ethologica*, 7(2), 103-108.
- Dziak, J. J., Dierker, L. C., & Abar, B. (2020). The interpretation of statistical power after the data have been gathered. *Current Psychology*, 39(3), 870-877.
- Hoening, J. M., & Heisey, D. M. (2001). The abuse of power: the pervasive fallacy of power calculations for data analysis. *The American Statistician*, 55(1), 19-24.
- Johnson, D. H. (1999). The insignificance of statistical significance testing. *The Journal of Wildlife Management*, 763-772.
- Taborsky, M. (2010). Sample size in the study of behaviour. *Ethology*, 116(3), 185-202.

Other references:

- Canteloup, C., Hoppitt, W., & van de Waal, E. (2020). Wild primates copy higher-ranked individuals in a social transmission experiment. *Nature Communications*, 11(1), 1-10.
- Farine, D. R., Spencer, K. A., & Boogert, N. J. (2015). Early-life stress triggers juvenile zebra finches to switch social learning strategies. *Current Biology*, 25(16), 2184-2188.
- Hasenjager, M. J., Hoppitt, W., & Leadbeater, E. (2020). Network-based diffusion analysis reveals context-specific dominance of dance communication in foraging honeybees. *Nature Communications*, 11(1), 1-9.
- Kline, R. B. (2004). *Beyond Significance Testing*. American Psychological Association, Washington, DC.

Liker, A., & Bókony, V. (2009). Larger groups are more successful in innovative problem solving in house sparrows. *Proceedings of the National Academy of Sciences*, 106(19), 7893-7898.

Luke, S. G. (2017). Evaluating significance in linear mixed-effects models in R. *Behavior Research Methods*, 49(4), 1494-1502.

Nakagawa, S., & Cuthill, I. C. (2007). Effect size, confidence interval and statistical significance: a practical guide for biologists. *Biological Reviews*, 82(4), 591-605.

1.3. At the same time, please address the rest of the reviewers' comments (noting that I agree with reviewer #2's suggestion that methodology in the supplementary material should be moved into the main text, although I'd keep the AIC tables where they are).

As also requested by Reviewer 2, we moved all text related to the applied methodology from the ESM to the Materials and Methods section of the revised ms (lines 96-309).

Responses to Reviewer 1's Comments

2.1. Small sample-size

For an experiment, I am surprised authors have considered only 16 individuals, 8 in a control group and other 8 eight with the *Spirulina* treatment. This reduces the power of study and increases the margin of error, mostly leading to Type 2 errors. I am aware it is frustrating to hear this after the experiments have been conducted, but this number is extremely small and if possible, I highly recommend authors to repeat the experiments with higher sample size, more replicas for the control and treatment groups.

We agree with the Reviewer that we had a rather modest sample size in the activity test as we observed one individual from each treatment group per each clutch, but we also expected a large effect of the applied developmental treatment based on the findings of similar studies (e.g., Chivers & Ferrari 2014). This expectation proved to be true as *Spirulina* treated tadpoles had a significantly lower activity compared to control individuals. However, the administration of the associative cue did not increase the activity of the *Spirulina* treated tadpoles to a similar extent, implying that tutor tadpoles did not associate “snail water” cues with the presence of *Spirulina* and thus did not increased their food searching activity when “snail water” was added to their water. In this particular case, the small sample size could indeed lead to Type II error (i.e., we conclude there is no significant effect, when actually there really is), but we can nevertheless claim that we did not found an effect of “snail water” that would have compensated for the effect of *Spirulina* supplementation.

In the foraging trials, we observed 3561 zone transition events performed by 128 tadpoles. Individuals were divided into two collective types, each type having eight replicates. We do not believe that this sample size would be particularly low compared to similar studies or inappropriate for statistical inference; please see also our rationale regarding this issue (and for studies with similar sample sizes) in our response to the Editor's Comment 1.2.

Chivers, D. P., & Ferrari, M. C. (2014). Social learning of predators by tadpoles: does food restriction alter the efficacy of tutors as information sources? *Animal Behaviour*, 89, 93-97.

2.2. Social information transmission

Authors consider the premise that social cue has been transmitted based on individuals' spatial decisions rather than on the nutrition per se of the spirulina. This seems to be a rather stretch of the question under study since the social cue under interest has not been transmitted in most of the cases. Further investigation should be conducted to address this question.

We agree with the Reviewer that this is also a possibility. However, we found that more tadpoles left the start zone (Z1), individuals exhibited a higher proportion of following during zone transitions, and tadpoles reached Z9 at a higher probability in the C collectives than in the S collectives. We argue that these differences between the two collective types cannot be explained by the presence of *Spirulina per se*. In line with these differences, NBDA clearly demonstrated that following interactions between tadpoles facilitated the discovery rate of the food patch only in the control collectives. Taken together, these findings indicated the influence of one another's movements on patch discovery in the untutored collectives, a phenomenon that was previously described in shoal-forming fish and termed as 'untransmitted social effect' (Atton et al. 2012, 2014). In the revised ms, we discussed our rationale regarding this issue more in detail (lines 400-413).

Atton, N., Hoppitt, W., Webster, M. M., Galef, B. G., & Laland, K. N. (2012). Information flow through threespine stickleback networks without social transmission. *Proceedings of the Royal Society B: Biological Sciences*, 279(1745), 4272-4278.

Atton, N., Galef, B. J., Hoppitt, W., Webster, M. M., & Laland, K. N. (2014). Familiarity affects social network structure and discovery of prey patch locations in foraging stickleback shoals. *Proceedings of the Royal Society B: Biological Sciences*, 281(1789), 20140579.

2.3. The effect of *Spirulina* on treated individuals

Authors defend in the discussion section (e.g., lines 296-299) that "individuals remained satiated and did not start exploring their environment when the associative cue was added to their water" to explain the lack of social transmission on the experiment group. This seems to be a rather baseline information to determine the timing of the experiment and the food used. Another question is: if *Spirulina* causes a decrease in the activity of the individuals, couldn't the authors use another food?

It is expected that adding extra food to tadpoles would decrease their activity, depending on its nutritional value. However, getting tutors familiar with the novel food type was necessary, otherwise these individuals could not have provide social cues to others related to the exploitability of the presented food patch. To counteract this expected effect of extra food, we used a neutral cue ("snail water") during the ontogenetic treatment that tutor tadpoles could associate with *Spirulina*. We expected that in the presence of "snail water" these conditioned tadpoles will search for food more actively at test; please also note that associative learning ability has been proven in tadpoles in various contexts in several frog species (e.g., Ferrari et al. 2010, Rothman et al. 2016) (lines 84-87). Contrary to our expectation, however, *Spirulina* treated individuals did not associate "snail water" cues with the presence of *Spirulina* and thus did not increased their food searching activity when "snail water" was added to their water (lines 350-356). Thus, the level of resource exploitation remained low in all tadpole collectives. Despite that, we found evidence that there were correlations between individual activity levels, discovery rates of a food patch (also by independent exploration), and the presence of social facilitation in patch discovery in the studied tadpole collectives (lines 400-414). As the presence of satiated and

therefore less active conspecifics among aggregating individuals can be considered a likely scenario in natural circumstances, we argue that conclusions drawn from this experiment may be extrapolated to foraging aggregations of other non-grouping animals as well.

Ferrari, M. C., Wisenden, B. D., & Chivers, D. P. (2010). Chemical ecology of predator–prey interactions in aquatic ecosystems: a review and prospectus. *Canadian Journal of Zoology*, 88(7), 698-724.

Rothman, G. R., Blackiston, D. J., & Levin, M. (2016). Color and intensity discrimination in *Xenopus laevis* tadpoles. *Animal Cognition*, 19(5), 911-919.

Responses to Reviewer 2's Comments

3.1. There is some scope for improving the presentation of the paper. Some of the terminology is confusing but this can easily be addressed with a revision. In particular, I was confused by the use of term 'knowledgeable' in the title and elsewhere. I took this at first to refer to the 'snail-conditioned water paired with spirulina' treatment groups, but in fact it refers to the tadpoles that have already visited the food once, irrespective of treatment group. This should be made clear early on. The snail + spirulina treatment confused me a bit- was this intended to lead to classical conditioning, so that conditioned tadpoles searched more actively at test? If so, did it work? Do you have the data to show this?

We thank the Reviewer for drawing our attention to the confusing terminology. Indeed, "knowledgeable" may refer to both *Spirulina* treated tadpoles and those that already discovered the food patch irrespective of their ontogenetic treatment. However, we believe that the emphasis should be put on the fact that tutors, individuals that had previous experience with the presented novel food type, failed to facilitate the exploitation of this resource in their collectives. In line with that, we changed the title of our manuscript to avoid further confusions. We are also grateful to the Reviewer for pointing out that conditioning, an important aspect of the experimental setup, was not emphasized sufficiently in the previous version. We included more information about the rationale behind its application into the revised ms and discussed the findings of the activity tests more in detail (lines 84-87, 350-356, 400-406, 436-454).

3.2. L42: Please define 'differentiated social structure'

We reworded this sentence in the revised ms (lines 42-44).

3.3. L42: Non-social is probably the wrong term since even animals that live alone are exposed to social cues and interactions. Non-grouping is perhaps a better term.

We agree with the Reviewer and replaced the term 'non-social' to 'non-grouping' throughout the revised ms.

3.4. L44: What is meant by 'temporal aggregations'?

We corrected this typo in the revised ms (line 44-45).

3.5. L49: Connections need not be random?

We aimed to propose that network connections can be based on incidental observation of others, and even these connections may predict the diffusion of the target behaviour through the network. We rephrased this part of the text to avoid unnecessary confusion in the revised ms (lines 50-52).

3.6. L92: Can you be sure if the tadpoles were non-kin if they were spawned in the wild? Could multiple egg clutches be fertilised by the same male?

Indeed, egg clutches may be fertilized by more than one male in common frogs, so there was some chance that certain tadpoles from different clutches could be related. On the other hand, the tested tadpole collectives consisted of one randomly selected individual per clutch, so the potential presence of kin might have affected the two types of collectives with an equal probability. We added this information to the revised ms (lines 99-104). We also examined in a supplementary analysis how the number of gates and number of conspecifics in the zone affected the time an individual spent in a zone (linear mixed-effects model, $n=3561$). We found no indication for the presence of social attraction (which was observed in a previous study between kin; Nicieza 1999) in the collectives (please see the figure below), or difference between collective types in this regard. Upon request, we can include this analysis into the main text or the ESM.

Nicieza, A. G. (1999). Context- dependent aggregation in Common Frog *Rana temporaria* tadpoles: influence of developmental stage, predation risk and social environment. *Functional Ecology*, 13(6), 852-858.

3.7. L96: Perhaps worth mentioning that the snail water was used as a conditioned stimulus first (I presume it was?), otherwise readers will be wondering why you added them. Can you provide more detail on the responses of tadpoles to this CS- do they respond, and if so, how? This needs to be demonstrated.

Thank you for pointing out this weakness. This is an important detail in applied experimental design, so we included more details about the administered conditional

stimulus as well and put more emphasis on discussing the findings of the activity tests in the revised ms (lines 84-87, 350-356, 436-461).

3.8. L124 & S1: Why the partitions? Why not use an open arena? Could the partitions have impeded movement? It certainly seems like they could have prevented the transmission of visual information between tadpoles.

We used a structured arena because we aimed to define potential leaders and followers; in an open area, we could not have determined which individual movement in the arena elicited a following response in another animal. Therefore, we divided the area into equal-size zones and assumed that individuals in the same zone can detect each others' movement through visual or other (e.g., acoustic or hydraulic) cues. Please note that this setup was necessary as, similarly to most model species (but see Strandburg-Peshkin et al. 2013, Rosenthal et al. 2015), there is no information about the visual detection range of common frog tadpoles. We added this argument to the revised ms (lines 155-158).

Rosenthal, S. B., Twomey, C. R., Hartnett, A. T., Wu, H. S., & Couzin, I. D. (2015). Revealing the hidden networks of interaction in mobile animal groups allows prediction of complex behavioral contagion. *Proceedings of the National Academy of Sciences*, 112(15), 4690-4695.

Strandburg-Peshkin, A., Twomey, C. R., Bode, N. W., Kao, A. B., Katz, Y., Ioannou, C. C., ... & Couzin, I. D. (2013). Visual sensory networks and effective information transfer in animal groups. *Current Biology*, 23(17), R709-R711.

3.9. L142-144: More information on how the leader-follower interaction was determined is needed here. Did you impose a cut off time, so that a follower event could only occur within a fixed time? Did the follower need to originate from the same compartment as the leader? Did the leader still need to be in the new compartment when the follower arrived?

We appreciate the suggestion. We added the requested information related to the definition of leader-follower interactions to the text (lines 241-246).

3.10. ESM: I think that all of the material in the ESM (except perhaps the tables) should be moved to the main text- this info is all important for understanding the procedures and analyses, and since RSOS has no word limit (as far as I'm aware anyway), there is no reason to separate the material between two different documents.

We moved all text related to the applied methodology from the ESM to the main text as suggested (lines 96-309).